# Electrochemical Characteristics of *Shewanella loihica* PV-4 on Reticulated Vitreous Carbon (RVC) with Different Potentials Applied

**DOI:** 10.3390/molecules27165330

**Published:** 2022-08-21

**Authors:** Shixin Wang, Xiaoming Zhang, Enrico Marsili

**Affiliations:** 1School of Science, Minzu University of China, Beijing 100081, China; 2Department of Chemical and Materials Engineering, School of Engineering and Digital Sciences, Nazarbayev University, Nur-Sultan 010000, Kazakhstan

**Keywords:** reticulated vitreous carbon (RVC), *Shewanella loihica* PV-4, biofilm formation, direct electron transfer (DET), mediated electron transfer (MET), bioelectrochemical systems (BES)

## Abstract

The current output of an anodic bioelectrochemical system (BES) depends upon the extracellular electron transfer (EET) rate from electricigens to the electrodes. Thus, investigation of EET mechanisms between electricigens and solid electrodes is essential. Here, reticulated vitreous carbon (RVC) electrodes are used to increase the surface available for biofilm formation of the known electricigen *Shewanella loihica* PV-4, which is limited in conventional flat electrodes. *S. loihica* PV-4 utilizes flavin-mediated EET at potential lower than the outer membrane cytochromes (OMC), while at higher potential, both direct electron transfer (DET) and mediated electron transfer (MET) contribute to the current output. Results show that high electrode potential favors cell attachment on RVC, which enhances the current output. DET is the prevailing mechanism in early biofilm, while the contribution of MET to current output increased as the biofilm matured. Electrochemical analysis under starvation shows that the mediators could be confined in the biofilm. The morphology of biofilm shows bacteria distributed on the top layer of honeycomb structures, preferentially on the flat areas. This study provides insights into the EET pathways of *S. loihica* PV-4 on porous RVC electrodes at different biofilm ages and different set potential, which is important for the design of real-world BES.

## 1. Introduction

Dissimilatory metal-reducing bacteria, such as *Shewanella* sp. and *Geobacter* sp., are capable of reducing insoluble extracellular electron acceptors such as metals and electrodes via microbially produced redox mediators, outer membrane cytochromes (OMC) and protein-based conductive appendages termed nanowires [1,2,3,4]. Because of this property, these microorganisms have been termed electricigens, indicating both the capacity to modulate their metabolism according to the redox potential of the electron acceptor [5] and the capability of converting chemical energy into electrical energy [6]. More recently, it has been observed that numerous bacterial and fungal species can transfer electrons to extracellular electron acceptors, via microbially produced redox mediators or when exogenous mediators are added. Differently from *Shewanella* sp. and *Geobacter* sp., the specific current output is also smaller, thus these microorganisms are termed weak electricigens, while *Shewanella* sp. and *Geobacter* sp. are often termed strong electricigens [7]. Electricigens have been used in numerous bioelectrochemical systems (BES), including microbial fuel cells (MFCs), bioelectrosynthesis and biosensors, both in controlled conditions and in situ [8,9,10,11,12,13]. The current output of these devices depends, among other factors, on the formation of biofilm, the concentration of low-conductivity extracellular polymeric substance (EPS) in the biofilm [14], the biofilm thickness and viability and the specific surface of the electrode (m^2^ m^−3^) that is available for biofilm formation and EET [15]. Research on electrode materials for BES has focused on low-cost materials, suitable for scale-up of laboratory devices [16,17,18]. Reticulated Vitreous Carbon (RVC) foam is a relatively impervious form of carbon, which is formed in a rigid three-dimensional (3D) honeycomb-like structure. The advantages of RVC as electrode material include easy preparation and handling, strong chemical and heat resistance, high specific surface and current densities, low electrical/fluid flow resistance and the ability to hold infused materials within controlled pore sizes. Since the early 1980s, researchers have reported the use of RVC as an electrode material for different categories of batteries, such as zinc-manganese dioxide cell, lead-acid batteries and lithium batteries [19,20,21,22,23,24,25]. As a 3D carbon material with tunable pore size, RVC is particularly suitable to support biofilm growth and cell confinement [23,25]. More recently, RVC was used to enhance microbial electrosynthesis of acetate in acetogenic mixed consortia [26]. Biofilm attachment and activity was further enhanced by synthesizing a nanopore network over commercially available RVC [27]. RVC electrodes have also been tested in the bioelectrochemical reduction of CO_2_ for methane production. However, only a few studies on RVC as electrode material for bioelectrochemical systems have been published Since biofilm is the prevalent phenotype in long-term operation of BES, a thorough understanding of electricigens biofilm formation is necessary [28]. Recent investigations found that the anode potential could regulate the growth and electrochemical activity of the microorganisms [29,30,31,32], and microbes might adjust their redox metabolism to adapt to the applied potentials, which can be used in electro-fermentation [30,31,32]. However, the combined effects of biofilm age, electrode potential and electrode material on the current output are still not well understood, which hinders full-scale applications of BES. A thorough understanding of these effects will enable a more rational optimization of BES for energy recovery, biosensing and other applications.

In this work, RVC was chosen as an electrode material to grow the model strong electricigen *Shewanella loihica* PV-4 biofilm under different electrode potential in three-electrode electrochemical cells. Results showed a strong effect of biofilm age and electrode potential on the EET process.

## 2. Results

### 2.1. Oxidation of RVC

After 50 electrochemical cycles in diluted acid solution (see Section 3.1), the activated RVC showed a stable CV curve, and the redox peak current for the test redox active compound K_3_Fe(CN)_6_ was approximately three times higher than the original RVC, demonstrating that the activation enhanced electron transfer rate (Figure 1). Also, the potential difference ΔE_p_ between the oxidation and reduction peak did not increase significantly, indicating that the RVC skeleton structure was not damaged by the acid oxidation treatment. High concentration of –OH and –COOH groups on the activated RVC surface also improved contact between *S. loihica* PV-4 OMC and the electrode surface [16]. Electrochemical activation is a low-energy method to oxidize RVC surface with respect to concentrated acid treatment, in which the modified RVC is no longer suitable for use as electrode in voltammetric experiments [33]. Moreover, the activated RVC had better stability within the potential window than the as-received RVC, and no systematic current output instability was observed in the electrochemical experiments (data not shown).

### 2.2. Potential Setup and Chronoamperometry (CA)

For *Shewanella* sp., it is well-known that OMCs and flavins are the main agents that transfer electrons to the acceptor (anode) in the direct and mediated EET pathways, respectively. EET from microorganisms to solid state acceptors can proceed only when the lowest energy level of the conduction band of the acceptor is lower than the redox potential of the electron-transferring agents. Previous investigations found that purified OmcA, MtrC, and CymA showed a broad potential range spanning approximately between −350 and 10 mV vs. standard calomel electrode (SCE), which is independent of the electrode material. However, in viable cells, due to continuous supply of electrons and the electrochemical resistance of the bacterial membrane, redox potential showed a large positive shift compared to the purified proteins [34]. The microbially produced flavins have a lower reduction potential, approximately −0.44 V vs. SCE at circumneutral pH [35]. Therefore, we separated the potential window into three zones as shown in Figure 2 (flavin zone, cytochromes zone and mixed transport zone) depending on the electrode potential.

Biofilm growth and the associated current output were studied in short-term experiments under well-defined conditions in which the effects of electron transfer reactions at the cathode were removed by using a potentiostat to fix the anode at a constant potential. Experiments at different electrode potentials were carried out to systematically investigate the EET process.

Following inoculation, there was a lag period prior to the onset of current output in the CA (Figure 3a). At electrode potential of −0.24 V, the current output was lower than 6 μA after 24 h and dropped to near zero after medium change (MC), indicating that the current output was contributed mostly by planktonic cells or suspended redox mediator, which is removed in the MC, rather than by the biofilm formed on the RVC surface. The electrode potential −0.24 V vs. SCE could not promote the mediated EET between bacteria and RVC electrode, even though the midpoint potential of the redox mediator flavins is −0.44 V (Figure 3b). This result further confirms that the contribution of biofilm to the overall current output is dominant. While early research suggested that riboflavin served as a soluble redox mediator in the EET process [35], more recent work proved that riboflavin must be attached to the bacterial membrane to promote efficient EET between cells and electrode [36]. When the working electrode potential was increased to 0 and 0.24 V, as shown in the red and green lines in Figure 3, respectively, the current output increased. These results were consistent with previous findings [37], where the EET rate was accelerated by increasing the working electrode potential. In an attempt to eliminate the electron acceptor limitation, the potential of 0.5 V was applied to the anode to maximize the current output. As shown by the blue line in Figure 3a, the current output increased to 350 μA in the first 24 h after inoculation, which was higher than the current output at 0 V (~50 μA) and 0.24 V (~100 μA), but it decreased sharply after MC. This phenomenon can by explained in two ways: (a) the planktonic cells contribute mostly to the current output or (b) the potential of 0.5 V damages the bacterial membrane and decreases biofilm viability, particularly in absence of planktonic cells that can attach to the surface and replace the damaged cells. While explanation (a) would indicate a radical change in the EET mechanism as electrode potential increases, explanation (b) is consistent with previous research, showing a negative effect of high electrode potential on current output [38]. Therefore, an applied electrode potential >−0.24 V and <0.5 V can modulate the current output of S. loihica PV-4 on RVC. Further, it appears that electrode potential higher than −0.24 V can favor cell attachment and biofilm formation, as demonstrated for other electricigens [39]. These results improved our understanding of the physiological adaptations required for biofilm growth on electrodes polarized at oxidative potentials.

Compared to other electrode materials, the optimal current output (at 0.24 V) at RVC was much higher [40] and gave a current density of 67.3 µA cm^−2^ after 96 h of growth and repeated MC to supply an adequate concentration of lactate as electron donor. It should be noted that the calculated electrode surface area is not the real surface area but rather the geometric area. This high current density should be attributed to RVC porosity, high hydrophilicity and roughness after the acid activation, which facilitate the adhesion of bacteria and biofilm formation.

In the CV, only one redox peak was observed at −0.24 V, which was centered at ~−0.37 V (Figure 3b). At electrode potential of 0 V and 0.24 V, there were two redox peaks, one centered at ~−0.46 V and the other centered at ~−0.29 V, indicating that at least two EET processes happened on the electrode surface. Based on the reduction potential analysis [34,35], we attributed these peaks at flavin and OMC, respectively, as previously reported for the high surface TiO_2_@TiN electrode [41]. At higher potential, the current increased further, particularly for biofilms grown on electrodes poised at 0.24 V due to the turnover conditions. However, no evident peak was observed, with the exception of a small anodic peak at E > 0.2 V for the electrode poised at −0.24 V and 0.24 V. These results confirm that both flavin-mediated EET and direct EET via OMC occurred at the same time on the electrode surface maintained at oxidation potential.

### 2.3. SEM Images

Only the biofilm grown on electrodes set at 0 V and 0.24 V were chosen for further experiments and biofilm characterization through SEM. As shown in Figure 4a,b, the average pore opening diameter of the RVC material was 50–200 µm, where *S. loihica* PV-4 cells adhered non-uniformly around the pores of the RVC skeleton. Representative SEM images of clean RVC are shown in Figure 4e,f. Biofilm formation did not alter the pore structure. The biofilm appears denser and more structured at 0.24 V (Figure 4c) than at 0 V (Figure 4d), which is consistent with the higher current output observed (Figure 3a).

Compared with the biofilm poised at 0.24 V (Figure 4c), the biofilm at 0 V was much thinner as shown in Figure 4d, with monolayer or sub-monolayer located on top surface. Thus, the activation of RVC electrode and the application of the optimal electrode potential increases the availability of adsorption sites at RVC surface, thus increasing biofilm formation and current output.

### 2.4. Electrochemical Characteristics of Biofilm

To further confirm the EET mechanism between biofilm and RVC electrode, the electrochemistry of early biofilm formed on electrodes set at 0.24 V following different stimuli was investigated. After a slow onset of current output in the first 20 h, likely due to the diffusional limitations discussed above, the CA (Figure 5a) shows a well-defined current output increase after riboflavin (0.5 µM). After addition of electron donor (lactate 20 mM), the current output increased rapidly and then kept increasing at a constant rate, indicating additional *S. loihica* PV-4 growth and biofilm formation. These results are different from those observed with 2D graphite electrodes, where the current increased rapidly a with high starting point because there are no strong diffusional limitations on a flat surface [42]. Therefore, the use of 3D RVC electrodes is not advantageous for short-term bioelectrochemical devices, (e.g., laboratory biosensors), but it is preferable for long-term BES applications, like environmental sensors and energy conversion.

This observation is confirmed by CV and DPV (Figure 5b,c). The small redox peaks observed immediately after inoculation were mainly from the riboflavin left in the planktonic bacteria, even though, prior to inoculation, the cells had been centrifuged and washed. In fact, the membrane-associated flavins may not be removed even after repeated washing. With time, the peak related to flavins increased little, while the peak at −0.15 V from OMC grew strongly, demonstrating the successful attachment and biofilm formation on RVC electrode surface.

The electrochemical characteristics of early biofilm under starvation, in which no lactate was provided, were also investigated. The resulting plateau in the CA curve, as shown in Figure 6a, revealed the successful biofilm growth on RVC. The CV (Figure 6b) and DPV (Figure 6c) also indicated that the current output was mainly due to DET via OMC, rather than from flavin-mediated EET.

While experiments on early biofilms are important to determine the biocompatibility of the electrode surface, information on the long-term functioning of BES is necessary for their real-world applications, and thus the EET mechanism of old biofilm was further investigated, as shown in Figure 7 and Figure 8.

During the experiment, no media change was carried out, however an additional 20 mM lactate was added about every 24 h. At early stage in biofilm growth (<48 h), OMC contributed mostly to EET, while at later time (72 and 96 h), biofilm-immobilized and dissolved flavins contributed mostly to the overall current output, eventually hindering the OMC peak in the CV and DPV.

When the spent medium was replaced with fresh medium without lactate (Figure 8), the current output dropped rapidly to near zero. However, a broad peak with large ΔE_p_ was observed, which increased and moved at lower potential after 23 h of starvation. It is unclear whether these peaks in the CV and DPV are related to flavins or release of intracellular enzymes from the starving *S. loihica* PV-4 cells (Figure 8b,c). Interestingly, the separation between oxidation peak and reduction peak increased as the scanning speed increased (Figure 8a, inset), however the trend is not monotone, suggesting that the underlying redox process was jointly diffusion- and surface-controlled.

To confirm the effect of mediators and planktonic cells on the current output, the fluorescence spectroscopy of spent medium at different biofilm age on graphite (Figure 9a) and RVC electrode (Figure 9b), respectively, was measured. The fluorescence of cytochromes is usually located between 400 and 440 nm [34], thus the peak at ~440 nm was attributed to the cytochromes of planktonic cells. As to the peak at 500–510 nm, it was attributed to riboflavin based on the known riboflavin spectrum, as confirmed by the sterile control experiment (Figure 9c).

To compare the relative intensity of cytochromes and riboflavin in medium suspension at different biofilm ages, we ran a multi-peaks analysis of the fluorescence spectroscopy, and the peak areas corresponding to cytochrome (440 nm) and riboflavin (510 nm) are listed in Table 1. At graphite electrodes, the ratio of cytochromes to riboflavin was >1 at 24 h after inoculation (AI) but decreased slightly after the first and second MC, confirming that MET contribute more than flavins to the EET process on graphite [42]. A similar trend was observed at the RVC electrode, however the cytochrome to riboflavin ratio was slightly higher after the first MC, indicating a slightly higher contribution of cytochromes to the EET process at RVC electrodes.

## 3. Material and Methods

### 3.1. Electrode Preparation

RVC material (500 PPI) was machine cut into 2 × 1 × 0.2 cm electrodes. Electrodes were treated via 50 cycles CV scanning in the range −0.2–1.15 V vs. SCE in 0.5 M H_2_SO_4_ solution at a scan rate of 250 mV s^−1^ and 10 cycles at 100 mV s^−1^, then soaked overnight in 1 M HCl to remove metals and other contaminants and stored in deionized water. The activation results were tested via 2.5 mM K_3_Fe(CN)_6_ in 1 M KNO_3_ by CV.

### 3.2. Assembly of Electrochemical Cells

RVC working electrodes were attached to 0.1 mm Pt wires (Sigma–Aldrich, St. Louis, MO, USA) via miniature nylon screws and stored in deionized water until use. Pt counter electrode wires were inserted into glass capillaries and soldered to copper wires. Reference electrodes were connected via a 3 mm glass capillary and Vycor frit. Cells equipped with three electrodes were autoclaved, and the salt bridge was filled with 0.1 M Na_2_SO_4_ in 1% agar. Autoclaved electrochemical cells were filled with sterile growth medium and operated under a flow of sterile humidified oxygen-free N_2_ at 30 °C and mixed with a magnetic stirrer. Sterile reactors were analyzed before each experiment by CV to verify the absence of redox active compounds. Electrochemical cells showing residual peaks in differential pulse voltammetry (DPV), anodic current in cyclic voltammetry (CV) or baseline noise were discarded as having possible electrode cleanliness or connection noise issues. The autoclaved, verified bioreactors were then used for growth of *S. loihica* PV-4 cultures.

### 3.3. Bacterial and Growth Medium

*S. loihica* PV-4 was grown aerobically for 24 h at 30 °C in modified Luria-Bertani medium (MLB) containing per liter: 10 g peptone, 5 g yeast extract, 10 g NaCl, 0.5 g Fe(III) citrate in deionized H_2_O. Subsequently, the culture was centrifuged, and the LB was replaced with 10 mL of modified defined media (MDM) containing per liter: 2.5 g NaHCO_3_, 0.08 g CaCl_2_·2H_2_O, 1.0 g NH_4_Cl, 0.2 g MgCl_2_·6H_2_O, 10 g NaCl, 7.2 g HEPES, 10 mL trace metal solution and 1 mL vitamin mix. The cells were further cultivated aerobically in MDM at 30 °C for 2 days using 15 mM lactate as a carbon source. The suspension was centrifuged for 15 min at 13,400 rpm, and the resultant pellet was washed with MDM three times before inoculation in the electrochemical cell. All chemicals used in this study were of reagent grade.

### 3.4. Growth in Electrochemical Cells

Electrochemical measures were performed with a VSP potentiostat (Bio-Logic, Grenoble, France). A constant potential was applied to working electrodes, and the biofilm formation was monitored by Chronoamperometry (CA), differential pulse voltammetry (DPV) and cyclic voltammetry (CV). Analyses were performed without stirring enabled. The parameters were: for DPV, E_i_ = −0.8 V, E_f_ = 0.3 V, pulse height = 50 mV, pulse width = 300 ms, step height = 2 mV, step time = 500 ms, scan rate = 4 mV s^−1^, current average over the last 80% of the step (1 s, 12 points), accumulation time 5 s; and for CV, equilibrium time 5 s, all scan rates = 1 mV s^−1^ except stated otherwise, E_i_ = −0.8 V, E_f_ = 0.3 V, current averaged over the whole step (1 s, 10 points). CA was carried out at different electrode potentials, as indicated. All potentials are vs. SCE. CV and DPV tests were performed regularly during the entire experimental run.

### 3.5. Fluorescence Spectroscopy

Spectrophotometric measurements were conducted using Cary Eclipse (Agilent, Santa Clara, CA, USA) spectrophotometers. The fluorescence emission spectra in the range of 365–800 nm were recorded with excitation at 360 nm; excitation and emission slit widths were 5 nm, with PMT voltage 600 V. Data collection and processing were performed by use of Cary Eclipse Software (Agilent, Santa Clara, USA).

### 3.6. Scanning Electron Microscopy (SEM) Sample Preparation

The electrodes were gently washed by dipping the electrode into sterilized deionized water to remove loosely attached cells and the excess medium, mainly salt. The washing was repeated three times before the biofilm was taken for SEM measurement. The electrodes with biofilm were then kept in autoclaved container to minimize further bacterial growth and air-dried. Microscopic images were obtained with a JEOL 5900 LV microscope (Jeol, Tokyo, Japan).

## 4. Conclusions

Understanding EET mechanisms in electricigen bacteria will provide useful information for the optimization and scale-up of BES. Due to its low price and its ability to fit in a more compact and modular reactor design, RVC is a good anodic material for BES and was chosen to study the EET from *S. loihica* PV-4 biofilm to the electrode at different set potentials.

The EET pathway as well as the main metabolites involved are regulated by the set electrode potential. Further, biofilm formation is part of the bacterial response to electrode potential. Initial biofilm formation was favored by potential higher than the OMC redox potential. However, too high potential inhibited cell attachment and biofilm formation on RVC electrodes.

Porous RVC could provide enough surface for biofilm growth and thus higher current output with respect to conventional graphite electrodes under the same conditions. In this study, the highest current density of 673 mA m^−2^ was achieved after 96 h from inoculation. At the early stage, direct EET dominated the current output. With the biofilm growing, more riboflavin was produced and ultimately overlapped the cytochrome peak, and the electron transfer pathway changed into riboflavin-dominated mediated EET.

## Figures and Tables

**Figure 1 molecules-27-05330-f001:**
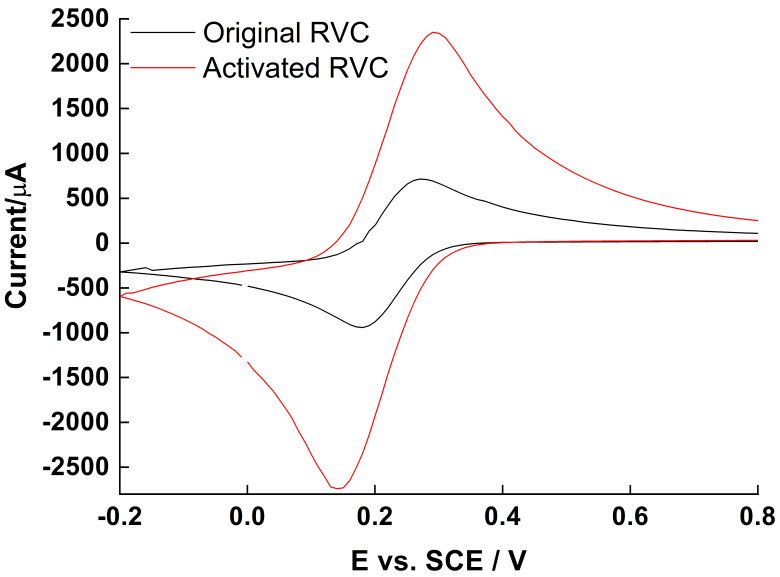
Cyclic voltammetry (CV) of 1.25 mM K_3_Fe(CN)_6_ in 1 M KNO_3_ on original and activated RVC electrode respectively, scan rate 10 mV s^−1^.

**Figure 2 molecules-27-05330-f002:**
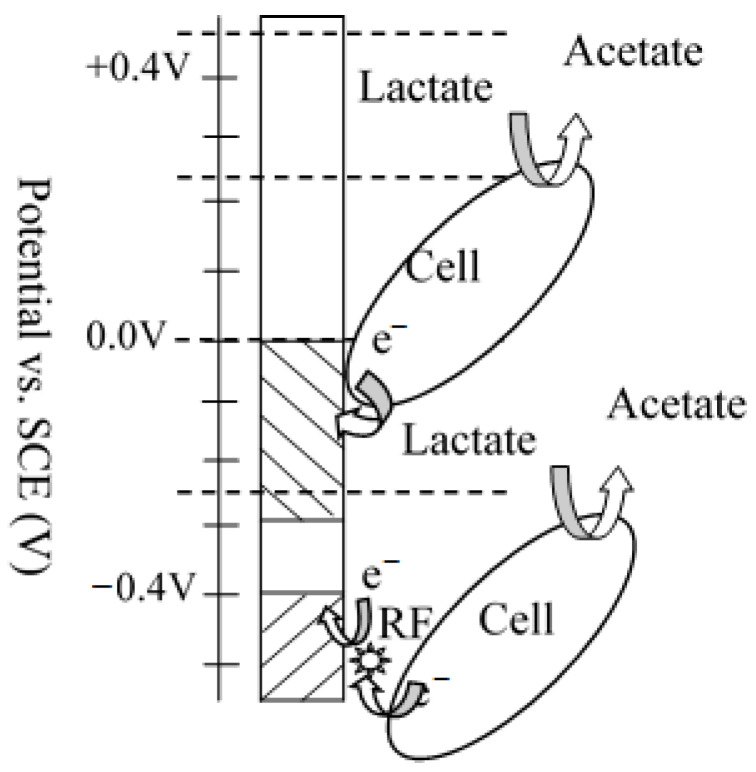
Cartoon diagram of shuttles EET process at different potential windows (mixed zone, cytochromes zone and mediator zones separated by dotted lines).

**Figure 3 molecules-27-05330-f003:**
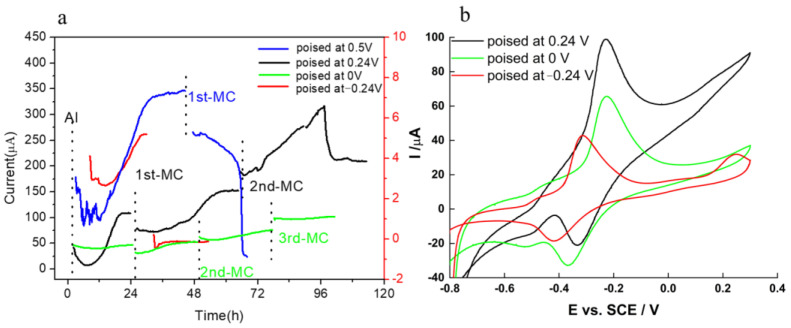
(**a**) CA of S. loihica PV-4 on RVC with working electrode potentials set at −0.24, 0, 0.24 and 0.5 V vs. SCE, respectively. At −0.24 V, the y-axis on the right was used to visualize the low current output; (**b**) CV of S. loihica PV-4 biofilm at 96 h after MC at different working electrode potentials. The CV of the experiment at 0.5 V vs. SCE did not show any peak and was not included in the figure. MC indicates the medium change, in which spent medium was replaced with fresh medium.

**Figure 4 molecules-27-05330-f004:**
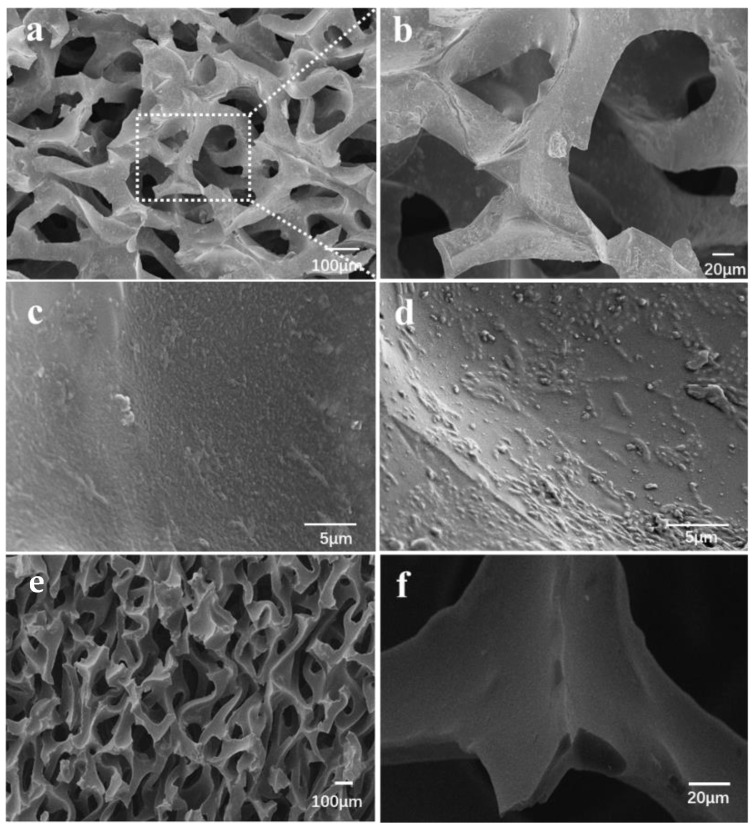
The distribution of *S. loihica* PV-4 biofilm on RVC electrode surface at 96 h after inoculation with electrode potentials 0.24 V (**a**–**c**) and 0 V (**d**). Biofilm formation is more evident at 0.24 V. Clean RVC at different magnifications is included for comparison (**e**,**f**).

**Figure 5 molecules-27-05330-f005:**
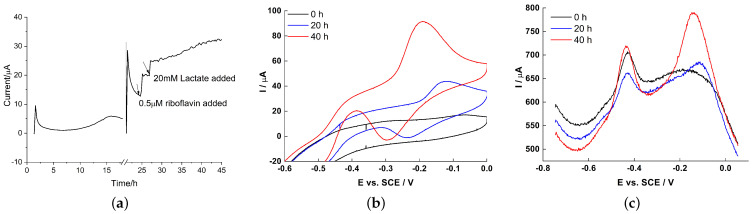
(**a**) CA at 0.24 V, (**b**) CV and (**c**) DPV of early *S. loihica* PV-4 biofilm grown on RVC.

**Figure 6 molecules-27-05330-f006:**
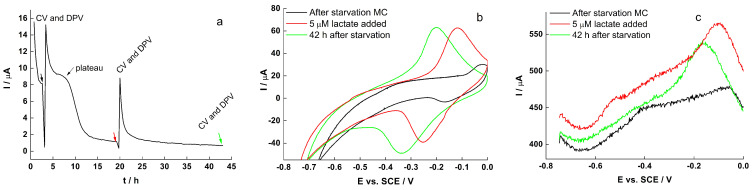
(**a**) CA, (**b**) CV and (**c**) DPV of early *S. loihica* PV-4 biofilm on RVC during starvation. The time t = 0 correspond to 48 h of growth at 0.24 V.

**Figure 7 molecules-27-05330-f007:**
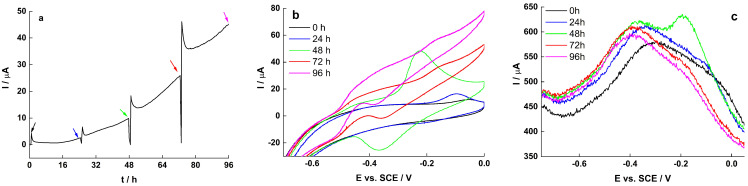
(**a**) CA, (**b**) CV and (**c**) DPV of *S. loihica* PV-4 biofilm on RVC. The color of the arrows in the CA corresponds to the color of the CV and DPV traces.

**Figure 8 molecules-27-05330-f008:**
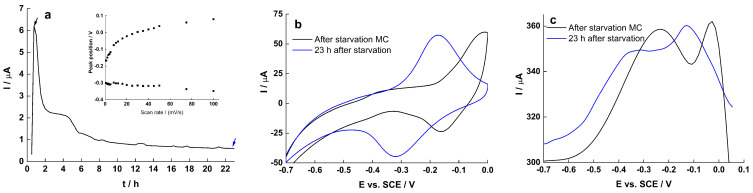
(**a**) CA with the relations of redox peak position vs. scan rate (inset); (**b**) CV and (**c**) DPV characteristics of early biofilm on RVC during starvation.

**Figure 9 molecules-27-05330-f009:**
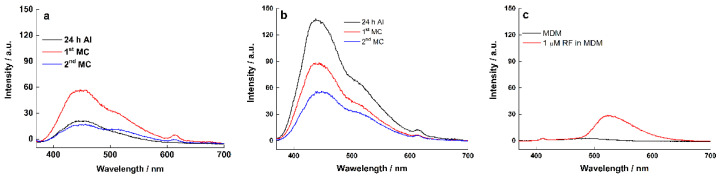
Fluorescence spectroscopy of medium suspension at different biofilm ages on (**a**) graphite electrode; (**b**) RVC electrode; (**c**) sterile control.

**Table 1 molecules-27-05330-t001:** Peak area of cytochromes and riboflavin at different biofilm ages.

Biofilm Age	Graphite Electrode (Peak Area)	RVC Electrode (Peak Area)	Graphite Electrode (Peak Area)	RVC Electrode (Peak Area)
Wavelength (nm)	440	440	510	510
24 h AI	2275	11,838	710	3372
1st MC	5304	7550	2126	1869
2nd MC	1847	4965	1038	1666

## Data Availability

Data presented in this work are not publicly available at this time but can be obtained upon reasonable request from the authors.

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
