# Peer review of "Electrochemical Characteristics of Shewanella loihica PV-4 on Reticulated Vitreous Carbon (RVC) with Different Potentials Applied"

_molecules, 2022, doi:10.3390/molecules27165330_

Round 1
Reviewer 1 Report
Reviewer Recommendation and Comments for manuscript molecules-1828919 with the title: “Electrochemical characteristics of Shewanella loihica PV-4 on reticulated vitreous carbon (RVC) with different potentials applied”, authors: S. Wang, X. Zhang, E. Marsili.
The authors present the studied the biofilm formed by S. loihica PV-4 on a vitreous carbon electrode at different potential values.
The article may be published after revision.
The main comments that I find useful for improving the quality of the article are presented below:
*line 159. ”At electrode potential of 0 V and 0.24 V, there were two redox peaks, one centered at -0.44 V and the other centered at -0.15 V, indicating that two EET processes happened on the electrode surface.” The second peak (-0.15 V) cannot be observed. It may be helpful to enter a voltamogram detail.
*Figure 3. In the case of CV poised on -0.24 V, other two anodic peaks are recorded (at -0.5 V and at 0.2 V). To which processes are these anodic peaks assigned?
*Figure 3. In the case of CVs poised on 0 V and 0.24 V, two cathodic peaks are recorded at -0.5 V. To which processes is this cathodic peak assigned?
*Figure 3b. For comparison, a CV must be entered, registered only for the growth medium (in the absence of S. loihica).
*The SEM image of the RVC electrode in the absence of biofilm (etalon) must be added in Figure 4.
*line 287. Relative peak areas unit measure?
*The typos must be corrected.
H2O, NaHCO3, CaCl2·2H2O, NH4Cl, MgCl2·6H2O. Use subscript.
Line 333. mV s-1
etc.
*The Molecules journal require a specific format of references, authors must pay more attention in their writing. No reference is written according to the format required by the journal.
*There are some grammar and typing mistakes.
*The authors must revise the entire manuscript.
Author Response
We thank the reviewer for his/her comments, which helped us improving the manuscript. In the following, the original reviewers’ comments are in bold, and the response in normal font. The changes to the original text, both here and in the revised manuscript are highlighted in yellow.
Reviewer Recommendation and Comments for manuscript molecules-1828919 with the title: “Electrochemical characteristics of Shewanella loihica PV-4 on reticulated vitreous carbon (RVC) with different potentials applied”, authors: S. Wang, X. Zhang, E. Marsili.
Reviewer #1:
The authors present the studied the biofilm formed by S. loihica PV-4 on a vitreous carbon electrode at different potential values. The article may be published after revision.
The main comments that I find useful for improving the quality of the article are presented below:
*line 159. ”At electrode potential of 0 V and 0.24 V, there were two redox peaks, one centered at -0.44 V and the other centered at -0.15 V, indicating that two EET processes happened on the electrode surface.”
The second peak (-0.15 V) cannot be observed. It may be helpful to enter a voltamogram detail.
The text has been corrected to reflect only the we-defined peaks observed in the voltammogram. Here is the revised text (line 172-182):
In the CV, only one redox peak was observed at -0.24 V, which was centered at ~-0.37 V (Figure 3b). At electrode potential of 0 V and 0.24 V, there were two redox peaks, one centered at ~-0.46 V and the other centered at ~-0.29 V, indicating that at least two EET processes happened on the electrode surface. Based on the reduction potential analysis [34, 35], we attributed these peaks at flavin and OMC, respectively, as previously reported for the high surface TiO2@TiN electrode [41]. At higher potential, the current increase further, particularly for biofilms grown on electrodes poised at 0.24 V, due to the turnover conditions. However, no evident peak was observed, with the exception of a small anodic peak at E > 0.2 V for the electrode poised at -0.24 V and 0.24 V. These results confirm that both flavin-mediated EET and direct EET via OMC occurred at the same time on the electrode surface maintained at oxidation potential.
Figure 3. In the case of CV poised on -0.24 V, other two anodic peaks are recorded (at -0.5 V and at 0.2 V). To which processes are these anodic peaks assigned?
We have revised the analysis of the CV results to account for all the peaks observed in the CV. See comment above and the related change of text.
Figure 3b. For comparison, a CV must be entered, registered only for the growth medium (in the absence of S. loihica).
The control CV of the blank medium in the absence of S. loihica can be found in Figure 5b (black trace – 0 h).
The SEM image of the RVC electrode in the absence of biofilm (etalon) must be added in Figure 4. The SEM images of clean RVC at two different magnifications were included as Figure 4e-f
line 287. Relative peak areas unit measure?
Apologies for the mistake. The caption has been changed accordingly. Fluorescence spectroscopy of the sterile medium has been included as Figure 9c.
Figure 9. Fuorescence spectroscopy of medium suspension at different biofilm ages on (a) graphite electrode; (b) RVC electrode; (c) sterile control.
The typos must be corrected.
H2O, NaHCO3, CaCl2·2H2O, NH4Cl, MgCl2·6H2O. Use subscript.
Line 333. mV s-1, etc.
All the typos indicated were corrected and highlighted in yellow, thank you.
The Molecules journal require a specific format of references, authors must pay more attention in their writing. No reference is written according to the format required by the journal.
We have checked thoroughly the References section and align it with the Journal guidelines.
There are some grammar and typing mistakes.
The manuscript was thoroughly revised for English grammar and typos.
The authors must revise the entire manuscript.
The entire manuscript was thoroughly revised according to the reviewers’ comments.
Reviewer 2 Report
In this paper the authors present an experimental activity aimed at testing reticulated vitreous carbon (RVC) utilization as an electrode in Microbial Electrochemical Systems by using Shewanella loihica PV-4 as a model organism. The authors investigated the mechanism at the basis of the electron transfer at different electrodes potentials, presence/absence of a source of energy, at different biofilm ages.
The paper is well written and organized. The methods are accurately described and justified when necessary. The obtained results support adequately the achieved conclusions.
I have just some minor remarks:
Line 53: The authors report the statement that previous studies focused on electricity production in planktonic cells while studying the electron transfer in MES. As it is written right now, it seems almost that that is a prevalent trend in studying electron transfer in MESs while, according to my knowledge, these are just part of a wider research field in which significant insights about electroactive biofilm formation, structure and behaviour have been given in relevant papers over time. Moreover, the papers the authors cite are quite outdated (2002 and 2010), leaving aside the scientific literature of, at least, the last 12 years. That being said, I think they should modify their statement making clear to the readers that PART of some (pioneeristic??) studies on electron transfer mechanisms focused their research activities on planktonic cells, but the role of biofilm on electricity production is more relevant.....etc
Lines 62-65: I think the authors should better describe the novel contribution of their work, in relation to the existing literature, with particular regard to the utilization of RVC in MES and the new insights about EET.
Line 79: The authors cite an "as-synthesized RVC": do they refer to untreated RVC? If not, they are required to explain what material they refer to.
Lines 184 - 194: this part seems somehow disjoined from the previous lines, where the authors justify the major development of biofilm on the electrode in consequence of the application of a -0.240 V potential. I can see that the microbial growth could be positively affected by a potential poised at -240 V (vs the calomelan reference electrode), but I can not see how the spear stress can justify the improved biofilm formation when biofilm grows at -0.240 V. In few words, the authors finish line 175 discussing the effect of potential but they start the following passage about the structure of electrode material with "This might be due to the fact...". Please check.
Maybe I missed it, but why in figure 9 the authors do not report the results of fluorescence spectroscopy for RVC electrode after the second medium changing?
References: almost an half of the references date back to more than 10 years. References to more updated works should be provided, even to better state the relevance of the presented research within the recent scientific literature in the same field.
Author Response
We thank the reviewers for his/her comments, which helped us improving the manuscript. In the following, the original reviewers’ comments are in bold, and the response in normal font. The changes to the original text, both here and in the revised manuscript are highlighted in yellow.
Reviewer 2#:
In this paper the authors present an experimental activity aimed at testing reticulated vitreous carbon (RVC) utilization as an electrode in Microbial Electrochemical Systems by using Shewanella loihica PV-4 as a model organism. The authors investigated the mechanism at the basis of the electron transfer at different electrodes potentials, presence/absence of a source of energy, at different biofilm ages.
The paper is well written and organized. The methods are accurately described and justified when necessary. The obtained results support adequately the achieved conclusions.
I have just some minor remarks:
Line 53: The authors report the statement that previous studies focused on electricity production in planktonic cells while studying the electron transfer in MES. As it is written right now, it seems almost that that is a prevalent trend in studying electron transfer in MESs while, according to my knowledge, these are just part of a wider research field in which significant insights about electroactive biofilm formation, structure and behaviour have been given in relevant papers over time.
Thank you for this comment. We have revised the introduction to include more recent studies about EET in biofilms and its applications into MES. Here is the additional text inserted:
Line 32-33
…via microbially produced redox mediators, outer membrane cytochromes (OMC) and pro-tein-based conductive appendages termed nanowires [1-4].
Line 36-42
More recently, it has been observed that numerous bacterial and fungal species can transfer electrons to extracellular electron acceptors, via microbially produced redox mediators or when exogenous mediators are added. Differently from Shewanella sp. and Geobacter sp., the specific current output is also smaller, thus these microorganisms are termed weak electricigens, while Shewanella sp. and Geobacter sp. are often termed strong electricigens [7].
Line 45-48
…on the formation of biofilm, the concentration of low-conductivity extracellular polymeric substance (EPS) in the biofilm, the biofilm thickness and viability and the specific surface of the electrode (m2 m-3) that is available for biofilm formation and EET.
Line 58-62
More recently, RVC was used to enhance microbial electrosynthesis of acetate in acetogenic mixed consortia [26]. Biofilm attachment and activity was further enhanced by synthesizing a nanopore network over commercially available RVC [27]. RVC electrodes have also been tested in the bioelectrochemical reduction of CO2 for methane production.
Line 75-76
Results showed a strong effect of biofilm age and electrode potential on the EET process.
Moreover, the papers the authors cite are quite outdated (2002 and 2010), leaving aside the scientific literature of, at least, the last 12 years. That being said, I think they should modify their statement making clear to the readers that PART of some (pioneeristic??) studies on electron transfer mechanisms focused their research activities on planktonic cells, but the role of biofilm on electricity production is more relevant.....etc
Thank you for your comment. We have thoroughly revised the introduction and discussion section to include more updated references, which account for the research progress in the characterization of biofilms on BES electrodes.
Lines 62-65: I think the authors should better describe the novel contribution of their work, in relation to the existing literature, with particular regard to the utilization of RVC in MES and the new insights about EET.
We have revised the introduction to frame the research contribution of the work reported. Basically, our results show the relative contribution to the overall current output of the two EET process in Shewanella loihica in the initial stages of biofilm formation.
More recently, RVC was used to enhance microbial electrosynthesis of acetate in acetogenic mixed consortia [26]. Biofilm attachment and activity was further enhanced by synthesizing a nanopore network over commercially available RVC [27]. RVC electrodes have also been tested in the bioelectrochemical reduction of CO2 for methane production.
Line 79: The authors cite an "as-synthesized RVC": do they refer to untreated RVC? If not, they are required to explain what material they refer to.
We have changed “as-synthesized” in “as-received” (Line 90)
Lines 184 - 194: this part seems somehow disjoined from the previous lines, where the authors justify the major development of biofilm on the electrode in consequence of the application of a -0.240 V potential. I can see that the microbial growth could be positively affected by a potential poised at -240 V (vs the calomelan reference electrode), but I cannot see how the spear stress can justify the improved biofilm formation when biofilm grows at -0.240 V. In few words, the authors finish line 175 discussing the effect of potential but they start the following passage about the structure of electrode material with "This might be due to the fact...". Please check.
Thank you for this comment. Since we have not studied in details the effect of shear stress on biofilm formation and electroactivity for RVC, we decided to remove the whole paragraph.
Maybe I missed it, but why in figure 9 the authors do not report the results of fluorescence spectroscopy for RVC electrode after the second medium changing?
We have included a different dataset for RVC electrode in Figure 9b, comprising also the fluorescence spectroscopy after the second medium changing. A sterile control (Figure 9c) has been also included. Peak areas were recalculated for all the datasets and included in Table 1.
References: almost an half of the references date back to more than 10 years. References to more updated works should be provided, even to better state the relevance of the presented research within the recent scientific literature in the same field.
We have revised the reference section and replaced older references with more recent ones.